# Sirtuin 3 Dependent and Independent Effects of NAD^+^ to Suppress Vascular Inflammation and Improve Endothelial Function in Mice

**DOI:** 10.3390/antiox11040706

**Published:** 2022-04-02

**Authors:** Xiaoyun Cao, Yalan Wu, Huiling Hong, Xiao Yu Tian

**Affiliations:** School of Biomedical Sciences, The Chinese University of Hong Kong, Hong Kong 999077, China; caoxiaoyun@link.cuhk.edu.hk (X.C.); yalanwu@cuhk.edu.hk (Y.W.); flora_hong@link.cuhk.edu.hk (H.H.)

**Keywords:** atherosclerosis, vascular inflammation, endothelial dysfunction, mitochondrial ROS, sirtuins

## Abstract

Atherosclerosis is initiated by endothelial cell dysfunction and vascular inflammation under the condition of hyperlipidemia. Sirtuin 3 (SIRT3) is a nicotinamide adenine dinucleotide (NAD^+^)-dependent mitochondrial deacetylase, which plays a key role in maintaining normal mitochondrial function. The present study tested whether endothelial-selective SIRT3 deletion accelerates vascular inflammation and oxidative stress, and assessed the protective effect of NAD^+^ to alleviate these changes in endothelial cells and in mouse models of atherosclerosis. We found that the selective deletion of SIRT3 in endothelial cells further impaired endothelium-dependent vasodilatation in the aorta treated with IL-1β, which was accompanied by upregulation of vascular inflammation markers and mitochondrial superoxide overproduction. Excepting the dysfunction of endothelium-dependent vasodilatation, such effects could be attenuated by treatment with NAD^+^. In human umbilical vein endothelial cells, SIRT3 silencing potentiated the induction of inflammatory factors by IL-1β, including VCAM-1, ICAM-1, and MCP1, and the impairment of mitochondrial respiration, both of which were alleviated by NAD^+^ treatment. In ApoE-deficient mice fed with a high-cholesterol diet, supplementation with nicotinamide riboside, the NAD^+^ precursor, reduced plaque formation, improved vascular function, and diminished vascular inflammation. Our results support the SIRT3-dependent and -independent of NAD^+^ to improve endothelial function in atherosclerosis.

## 1. Introduction

SIRT3 is a NAD^+^-dependent deacetylase that is mainly located in mitochondria, which regulates many mitochondrial proteins by post-translational modification [1,2,3,4]. It was first identified as yeast *SIR2*, equivalent to SIRT3, possessing NAD^+^-dependent protein deacetylase activity [1]. In later studies, using purified active SIRT3 and a histone-based peptide as substrates, the NAD^+^-dependent deacetylase activity of human SIRT3 was found in vitro [2,3], showing that the activities of mammalian SIRT3 proteins are also NAD^+^ dependent. The basic function of SIRT3 is to activate or deactivate mitochondrial target proteins by deacetylating the lysine residues. Through this action, SIRT3 regulates many mitochondrial proteins involved in the regulation of energy homeostasis [5], mitochondrial function [6], and lipid metabolism [7]. Studies show that SIRT3 is highly expressed in the most metabolically active tissues, such as liver, kidney, and heart [5,8]. Proteomic analysis of liver mitochondrial proteins indicated that the mitochondrial proteome is extensively acetylated in *Sirt3*^−/−^ mice [9]. Therefore, the role of SIRT3 in the liver and other metabolically active organs has been extensively explored.

In the cardiovascular system, SIRT3 also plays an important role, which is better demonstrated in cardiovascular disease. *Sirt3*^−/−^ mice have more vascular inflammation, increased oxidative stress, and high blood pressure [10], accompanied by an early vascular aging phenotype and shortened lifespan [11]. *Sirt3*^−/−^ mice also develop accelerated diet-induced obesity, insulin resistance, and hyperlipidemia [12]. In addition, SIRT3 attenuated diabetic cardiomyopathy via regulating TIGAR expression and improving cardiomyocyte metabolism [13]. Importantly, the SIRT3 inhibition associated with dysregulation of cardiovascular function and an accelerated vascular aging phenotype is generally coupled with decreased NAD^+^ levels. For example, the expression of SIRT3 and NAD^+^ levels decreases with age in sedentary adults compared to younger individuals [14,15]. In addition, the expression of SIRT3 in the heart of db/db diabetic mice and the tissue NAD^+^ levels in these mice are low compared to control non-diabetic *db*^+/−^ mice, which is associated with impaired angiogenesis [16,17]. Chronic high-fat-diet feeding results in reduced hepatic SIRT3 activity and NAD^+^ levels [12,18,19]. Therefore, strategies aimed at increasing intracellular NAD^+^ levels are now used as a nutritional supplement in humans. As an important mediator of NAD^+^ dependent effects, whether loss of SIRT3 attenuates some of the beneficial effects of NAD^+^ is not entirely clear.

The vascular endothelium is a direct target of many major cardiovascular risk factors. Vascular endothelial dysfunction is an important initial step for atherogenesis [20]. Accumulating evidence indicates the important role of SIRT3 in protecting endothelial function, which is mostly focused on its inhibition of oxidative stress. Overexpression of SIRT3 in mice protects against endothelial dysfunction and attenuates vascular oxidative stress in experimental hypertensive mouse models [10]. SIRT3 enhances reactive oxygen species (ROS) clearance via altering the acetylation level and subsequently activating mitochondrial antioxidant enzyme SOD2 [21,22,23], which reduces ROS by catalyzing superoxide (O_2_^−^) to hydrogen peroxide (H_2_O_2_). This SOD2 mediated effect of SIRT3 is observed not only in vascular tissues but also in other organs. For example, the liver from *Sirt3*^−/−^ mice exhibits decreased SOD2 activity [22]. Apart from the direct regulation of SOD2, SIRT3 can also activate the transcription factor FOXO3a, which induces the expression of its target gene *Sod2*, and catalase in an NAD^+^-dependent manner [24,25]. In addition to SOD2, SIRT3 activates the isocitrate dehydrogenase 2 (IDH2) pathway to suppress ROS production during increased oxidative metabolism, because IDH2 is a critical component of the mitochondrial antioxidant pathway through its ability to enhance the mitochondrial glutathione antioxidant defense system [26,27]. In fact, several drugs or small molecules act on SIRT3 to improve endothelial function. For example, rosuvastatin attenuates oxidative injury in endothelial cells (ECs) through maintaining mitochondrial ROS homeostasis, which is mediated by SIRT3 [28]. NaHS, which acts similarly to H_2_S, also improves mitochondrial function and inhibits oxidative stress in cardiomyocytes through activating SIRT3 in a mouse model of myocardial hypertrophy [29].

Previous studies suggested that increasing the endogenous NAD^+^ level by supplementing NAD^+^ precursors or intermediates has anti-atherogenic effects in a mouse model of experimental atherosclerosis and may also modulate cholesterol levels. For example, supplementation of nicotinamide mononucleotide (NMN), a key NAD^+^ precursor, in a mouse diet improves endothelium-dependent vasodilatation and reverses age-associated arterial dysfunction and oxidative stress [30]. The intracellular level of NAD is generated by the conversion from NMN by enzymes including NAMPT, CD73, and CD38. Supplementation of both nicotinamide riboside (NR), the precursor of NMN, and NMN, increases plasma and intracellular NAD^+^ partly through the conversion of extracellular CD73 [31]. NAD^+^ subsequently activates SIRT3, thus inhibiting ROS overproduction and reducing cholesterol levels in C57BL/6 mice fed with a high-fat diet [18]. Another precursor nicotinic acid (NA) supplement also showed an anti-atherogenic effect in hyperlipidemic *ApoE*^−/−^ mice at an advanced lesion stage [32]. In addition, NAM supplementation also prevents the development of atherosclerosis and inhibits inflammation in the aortas of *ApoE*^−/−^ mice [33]. NAM is both the metabolite of NAD^+^ and replenishes NAD^+^ through the NAMPT-dependent salvage pathway. Evidence also shows that NAMPT expression is protective against neutrophil infiltration and inflammation in atherosclerotic plaques, using NAMPT inhibitor FK866 [34]. NAM is required for the activity of nicotinamide-*N*-methyltransferase, which is an enzyme that is highly expressed in ECs for protection against oxidant injury [35]. In addition, NAM can be converted from NAD^+^ through NAD-consuming enzymes, including sirtuins, poly-ADP-ribose polymerase (PARPs), and ectoenzymes such as CD38 [36]. For instance, inhibition of PARP1 activity, which prevents PARP-dependent NAD^+^ degradation, promoted endothelial repair in a rabbit model of high-fat diet-induced atherosclerosis [37]. However, it is unclear whether sirtuins, such as SIRT3 in ECs, which have a known role against endothelial dysfunction and vascular inflammation, are required for the beneficial effect of NAD^+^ in endothelial cells and atherosclerosis.

To address the above questions, we aimed to examine the involvement of SIRT3 in the beneficial effect of NAD^+^ against vascular inflammation, the associated endothelial dysfunction, and ROS production in ECs and in hyperlipidemic *ApoE*^−/−^ mice.

## 2. Materials and Methods

### 2.1. Animals Experiments

All mice were kept at a constant temperature (22 ± 1 °C) under a 12 h light/dark cycle and had free access to water and standard chow unless specified elsewhere. Male apolipoprotein E deficient (*ApoE*^−/−^) mice at 8–10 weeks old were supplied by the Chinese University of Hong Kong (CUHK) Laboratory Animal Service Center (Hong Kong, China). The *Sirt3* floxed mutant mice (*Sirt3^tm1.1Auw^* or *Sirt3^L2^* were from Johan Auwerx Lab) and the endothelium-selective Cdh5-cre transgenic mice (*B6*;*129^-Tg^*
^(*Cdh5-cre*)*1Spe/J*^) were from Jackson lab. Both strains were backcrossed with C57BL/6 mice before they were crossed to generate EC selective deletion of *Sirt3* as *Sirt3^f/f^; Cdh5^Cre/+^* (Sirt3^EC-KO^) mice. Littermate wild-type mice *Sirt3^f/f^* (Sirt3^EC-WT^) were used as controls. All *ApoE*^−/−^ were fed a high-cholesterol diet (HCD, D12336, Research Diet) for 8 weeks to induce atherosclerosis. After that, mice were randomly divided into the vehicle group (distilled water) or NR group (400 mg/kg/day). Vehicle or NR was delivered by oral gavage for another 4 weeks, while the mice were maintained on HCD. All animal experiments followed the guideline of the Laboratory Animal Experimentation Ethical Committee of CUHK (AEEC: 19-125-GRF). All the animal experiments followed the ARRIVE guidelines.

### 2.2. Cell Culture and RNA Interference

Mouse brain microvascular endothelial cells (mBMECs, from Angio-Proteomie) were cultured in DMEM/HG medium (ThermoFisher, Waltham, MA, USA) supplemented with 20% FBS (ThermoFisher, Waltham, MA, USA), and 1% Antibiotic–Antimycotic (100×) solution (ThermoFisher, Waltham, MA, USA). Cells were infected with lentivirus carrying three short hairpin RNA (shRNA) sequences against mouse Sirt3 or a scramble lentivirus and selected with puromycin for stable expression. Puromycin was added 3 days after infection, with a concentration of 1 μg/mL, which was replaced every 2–4 days depending on the confluency of the cells. Human umbilical vein endothelial cells (HUVECs, from Lonza, Visp, Switzerland) between passages 6–9 were cultured in DMEM/F12 medium supplemented with 10% FBS, 1% Antibiotics–Antimycotics, and endothelial cell growth supplement (Sigma-Aldrich, St. Louis, MO, USA). HUVECs were transfected with human SIRT3 siRNA (GenePharma, Suzhou, China) or universal scrambled negative control siRNA using Lipofectamine RNAiMAX Transfection Reagent (ThermoFisher, Waltham, MA, USA), according to the manufacturer’s instructions and were used within 36–48 h after transfection.

### 2.3. Quantitative RT-PCR Analysis

Tissue or cells were lysed in RNAiso plus (Takara Bio Inc., Beijing, China) for total RNA extraction. RNA purity and concentration were measured by a Nanodrop spectrophotometer (ThermoFisher). A total of 1 µg RNA was reverse transcribed into cDNA using PrimeScript RT Master Mix (TaKaRa, RR036A, Beijing, China) following the manufacturer’s instructions. The mRNA expression levels of target genes were determined by quantitative RT-PCR with TB Green^®^ Premix Ex Taq™ (Takara, RR420A, Beijing, China) on the ViiA real-time PCR system (Applied Biosystems, Waltham, MA, USA). Primer sequences are listed in Table 1.

### 2.4. Western Blot

Cells or tissues were lysed in RIPA buffer with Protease Inhibitor Cocktail and PhosSTOP (both from Roche, Basel, Switzerland) for protein extraction. The lysates were incubated on ice for 30 min and then centrifuged for 15 min at 16,000× *g* and the protein content was measured using a BCA Protein Assay Kit (Thermo Fisher, 23225). Protein samples (10 or 20 μg) were mixed in loading buffer with 5% β-mercaptoethanol and denatured at 95 °C for 5 min. Proteins were separated by SDS-PAGE and transferred onto the PVDF membrane (Millipore, Burlington, MA, USA). The PVDF membrane was then washed with Tris-buffered saline with Tween-20 (TBST) and then blocked with 5% BSA in TBST for 0.5 h, followed by overnight incubation with primary antibodies including: anti-VCAM1 (1:1000, Abcam, ab134047, Cambridge, UK), anti-ICAM1 (1:500, Santa Cruz, SC-107, Dallas, TX, USA), anti-GAPDH (1:2000, Affinity biosciences, AF7021, Beijing, China) overnight at 4 °C. The membrane was washed with TBST and incubated with secondary goat anti-rabbit or anti-mouse IgG (H + L)-HRP conjugated antibodies (1:2000) for 1 h at room temperature. Band intensity was detected using ECL reagent, imaged, and quantified using the Bio-Rad ChemiDoc™ Imaging System. (Bio-Rad Laboratories, Hercules, CA, USA) 

### 2.5. Seahorse Test

Oxygen consumption rate (OCR) was measured using Seahorse XF96 Analyzers (Agilent, Santa Clara, CA, USA) according to the manufacturer’s protocol. Prior to the start of the experiment, cells were seeded (8000 cells/well) into the XF96 cell culture plate and allowed to adhere for 24 h. Cell culture media were replaced with XF media (Seahorse Bioscience, North Billerica, MA, USA) supplemented with 2 mmol/L Glutamax, 1 mmol/L sodium pyruvate, and 25 mmol/L glucose and were incubated for 1 h in a 37 °C CO_2_ incubator. Following the establishment of a basal OCR reading, the following pharmacological manipulators of mitochondrial respiratory chain proteins were injected sequentially: (i) oligomycin (1 μmol/L) at ATP synthase inhibitor; (ii) carbonyl cyanide-4-(trifluoromethoxy) phenyl hydrazone (FCCP) (1.5 μmol/L), as a mitochondrial uncoupler; followed by (iii) antimycin A (10 μmol/L), as a complex III inhibitor. The results were calculated using the Seahorse XF Mito Stress Test Report Generator (Seahorse Bioscience, North Billerica, MA, USA).

### 2.6. Vascular Reactivity

Mice were euthanized by CO_2_ anesthesia. Thoracic aortas were dissected in ice-cold oxygenated Krebs solution that contained the following composition (mmol/L): 119 NaCl, 4.7 KCl, 2.5 CaCl_2_, 1 MgCl_2_, 25 NaHCO_3_, 1.2 KH_2_PO_4_, and D-glucose, pH 7.4. Aortic segments at 1.5–2 mm were dissected and mounted in a wire myograph system (Danish Myo Technology, Aarhus, Denmark). The aortic segments were stretched to a baseline tension at 3 mN and then equilibrated for 60 min at 37 °C. Vessel viability was assessed by the response to 60 mmol/L KCl. Endothelium-dependent relaxation (EDR) was measured by testing the concentration-dependent responses to the cumulative addition of acetylcholine (ACh) in phenylephrine (Phe, 3 μmol/L)-precontracted aortic segments. Endothelium-independent relaxation to sodium nitroprusside (SNP) was performed in aortic segments after incubation with L-NAME (0.1 mmol/L) for 30 min to block nitric oxide synthase.

### 2.7. MitoSOX Staining

MitoSOX Red indicator (Invitrogen, M36008, Waltham, MA, USA) was used to detect mitochondrial superoxide generation in the en face preparation of the aorta. Briefly, aortic segments were cut open and incubated with MitoSOX (5 μmol/L) at 37 °C for 20 min. The fluorescent signal was visualized on an Olympus FV1200 confocal microscope (Olympus Corporation, Tokyo, Japan). A mean value of the MitoSOX fluorescence of three random images taken from each aortic segment was used as one measurement of each sample.

### 2.8. Histological Staining

Mouse heart was frozen in an OCT compound and cut at 10 µm as serial sections starting from the location when the aortic valve became visible. The section with the largest aortic root area with three visible valves was used for the representative image and analysis. Sections were washed and fixed in 4% paraformaldehyde. Oil Red O staining was conducted to detect lipid deposition of the aortic root section. Picrosirius red staining was used to examine collagen deposition in atherosclerotic lesions. Images were taken at 4× magnification under a bright field and polarized light. For en face Oil Red O staining of the whole aorta, aortas were removed and placed in 4% formaldehyde. The aorta was opened longitudinally from the heart to the iliac arteries and pinned out on a dissection plate. Fixed aortas were stained with Oil Red O to observe the lipid-rich plaque area. The Oil Red O-positive area in proportion to the total aortic surface area was measured and analyzed using Image J (National Institutes of Health, Bethesda, MD, USA). 

### 2.9. Immunofluorescence Staining

The cryosections of the aortic root were fixed in 4% paraformaldehyde and permeabilized using 0.1% Triton X-100 (Sigma-Aldrich, St. Louis, MO, USA). The sections were blocked using 5% normal goat or donkey serum (Abcam) at room temperature for 2 h and then incubated with primary antibodies, including anti-CD68 (1:100, Abcam, ab31630), anti-VCAM-1 (1:100, Abcam, ab134047), anti-E-Selectin (1:100, Santa Cruz, sc-137054), overnight at 4 °C followed by appropriate fluorescence-conjugated secondary antibodies (ThermoFisher, Waltham, MA, USA) for 2 h at room temperature in the dark. Nuclei were stained with Hoechst 33342 (ThermoFisher, Waltham, MA, USA) and mounted with a fluorescence mounting medium (Electron Microscopy Sciences, Cat#17985-10, Hatfield, PA, USA). Images were acquired using an Olympus FV1200 confocal microscope (Olympus Corporation, Tokyo, Japan). Quantification of immunofluorescence staining was performed with the Image J software. 

### 2.10. Serum Chemistry Analysis 

Animal blood samples were collected in tubes containing a clot activator and serum gel separator (BD Microtainer, 365967, Haryana, India) and centrifuged at 3000 rpm for 15 min at 4 °C. The serum supernatant was harvested. Total cholesterol (TC) concentration in the serum was measured using a cholesterol measurement kit (EKF stanbio, 1010-225, Barleben, Germany) according to the manufacturer’s instructions. Serum triglycerides (TA) concentration was measured using a triglyceride measurement kit (EKF stanbio, 2200-225, Barleben, Germany) according to the manufacturer’s instructions.

### 2.11. Statistical Analysis 

All quantitative results are presented as means ± SEM unless specified. The GraphPad Prism software (Version 6.0) (San Diego, CA, USA) was used for statistical analysis. Student’s *t*-test was used for comparison between two groups, while one-way ANOVA and multiple comparison tests were used for more than two groups. Moreover, * *p* < 0.05, ** *p* < 0.01, and *** *p* < 0.001 were used to indicate statistical significance. All the image analyses were performed in a single-blind manner and assigned to individual samples afterward.

## 3. Results

### 3.1. NAD^+^ Attenuated Inflammatory Response in Endothelial Cells in a SIRT3-Independent Manner

To understand the contribution of SIRT3 in mediating the anti-inflammatory effect of NAD^+^ in ECs, we first measured the protein and mRNA levels of EC genes involved in vascular inflammation. In HUVECs, 10 ng/mL of IL-1β was able to induce the expressions of VCAM-1 and ICAM-1, which are adhesion molecules expressed by activated ECs for immune cell adhesion (Figure 1A). The effect of IL-1β was attenuated by co-treatment with 1 mmol/L NAD^+^, while the effect of SIRT3 inhibitor (3-TYP, 100 µmol/L) to reverse the effect of NAD^+^ was only obvious with VCAM-1 expression (Figure 1A). Considering that 3-TYP might not be specific for SIRT3, we made a cell line using mBMECs with stable expression of either scramble or Sirt3 shRNA (Figure 1B). In Sirt3 shRNA-expressing mBMECs, the basal expressions of Icam1 and Ccl2 (also named as *MCP-1*, which is a chemoattractant that is highly expressed by activated ECs) were higher than scramble shRNA-expressing mBMECs (Figure 1C,D). Both IL-1β and TNFα were able to further upregulate both Icam1 and Ccl2, while the effect of TNFα was more obvious (Figure 1C,D). Therefore, we further used TNFα to test whether the effect of NAD^+^ was mediated by SIRT3 expression. However, co-treatment with NAD^+^ was able to suppress Vcam1 and Ccl2 expressions induced by TNFα even in Sirt3 knockdown ECs, indicating that the anti-inflammatory effect of NAD^+^ was probably independent of SIRT3.

### 3.2. NAD^+^ and SIRT3 Independently Improve Mitochondrial Function

On the basis of the well-known function of NAD^+^ and SIRT3 in regulating mitochondrial function, which is also related to vascular inflammation, next, we examined mitochondrial respiration in HUVECs. The effect of SIRT3 inhibition by siRNA at the basal level was significant in ATP-linked respiration, but not basal or maximal respiration (Figure 2A–E). The effect of IL-1β to suppress mitochondrial respiration was not significant in control (Scr) HUVECs (Figure 2A–E), whereas, in si-SIRT3-transfected HUVECs, IL-1β was able to suppress ATP-linked respiration (Figure 2E), suggesting that inhibition of SIRT3 sensitized ECs to the effect of IL-1β on inhibiting mitochondrial function. In addition, co-treatment with NAD^+^ was able to enhance basal, maximal, and ATP-linked respiration in both control and SIRT3 knockdown cells. NAD^+^ was also able to reverse the effect of IL-1β (Figure 2C–E). These results indicated that, although SIRT3 inhibition impaired mitochondrial function in the presence of a stressor, NAD^+^ was still able to boost mitochondrial function and increase the reserve bioenergetic capacity of ECs under both homeostasis and stress, irrespective of Sirt3 expression.

### 3.3. SIRT3 Deletion Enhanced Oxidative Stress in Endothelial Cells Attenuated by NAD

Mitochondrial dysfunction is closely associated with the production of reactive oxygen species (ROS), which induces oxidative stress-associated cellular damage. Alterations to respiration or the mitochondrial coupling capacity may disrupt the balance between the production and scavenging of mitochondrial ROS (mitoROS). We thus examined the effect of NAD^+^ on mitoROS production in the en face preparation of the aorta. MitoSOX was used to measure mitochondrial superoxide production, which was imaged by fluorescence from the aortic endothelium (Figure 3A). The results showed that similar to mitochondrial function, the basal level of mitoROS was moderately but not significantly increased in the endothelium from the endothelial selective *Sirt3* knockout (Sirt3^EC-KO^) mice compared to the wild-type (Sirt3^EC-WT^) mice (Figure 3A,B). Meanwhile, both 10 ng/mL of IL-1β and 50 μg/mL of oxidized LDL (ox-LDL, a major cardiovascular risk factor, which is able to induce oxidative stress) were able to increase mitoROS in the aorta of both genotypes. Knockout of *Sirt3* expression in Sirt3^EC-KO^ mice further enhanced mitoROS production induced by IL-1β and oxLDL. Importantly, similar to the effect on mitochondrial respiration, NAD^+^ suppressed IL-1β or oxLDL-induced mitoROS production with or without Sirt3 expression (Figure 3B), suggesting a SIRT3-independent effect of NAD^+^ to rescue mitochondrial function and reduce mitochondrial oxidative stress.

### 3.4. NAD^+^ Improved Vascular Function in a SIRT3-Dependent Manner

On the basis of the anti-inflammatory and antioxidant effects of NAD^+^ on ECs, which are independent of SIRT3, we next explored whether NAD^+^ alleviates vascular endothelial dysfunction in a similar manner because oxidative stress is closely associated with endothelial dysfunction. We used a vascular reactivity test to measure endothelium-dependent relaxation (EDR) in the mouse aorta in response to ACh, as an indication of endothelial function. Moreover, endothelium-independent relaxation in response to nitric oxide donor SNP was also measured to verify that vascular smooth muscle function was unaffected. The aortic rings were isolated from Sirt3^EC-WT^ and Sirt3^EC-KO^ mice. IL-1β similarly impaired EDR (an ~80% reduction, indicated by an open triangle in Figure 4A,B) in the aortic rings of both the Sirt3^EC-WT^ and Sirt3^EC-KO^ mice. NAD^+^ alone (indicated by the open square) did not show any effect in the aortic rings of either genotype. Co-treatment of NAD^+^ with IL-1β partially improved EDR in Sirt3^EC-WT^ mice (Figure 4A), but not in Sirt3^EC-KO^ mice (Figure 4B). SNP-induced relaxation was similar among all the groups (Figure 4C,D). These results suggested that the effect of NAD^+^ in improving endothelial function was, at least partially, dependent on SIRT3 expression.

### 3.5. Nicotinamide Riboside Reduced Atherosclerotic Plaque Formation in ApoE^−/−^ Mice

On the basis of the evidence that NAD^+^ improved endothelial function after impairment by IL-1β in the isolated aorta, we further examined the effect of increasing NAD^+^ concentration in vivo against endothelial dysfunction and vascular inflammation in a mouse model of atherosclerosis in vivo. *ApoE*^−/−^ mice were fed an HCD for 12 weeks. In the final 4 weeks, nicotinamide riboside (NR), the precursor of NAD^+^, was given to the mice by oral gavage at 400 mg/kg/day (Figure 5A). NR is a salvageable precursor of NAD^+^, which is known to increase NAD^+^ levels in various mammalian cell lines [38]. After NR treatment, both total cholesterol and triglyceride levels were reduced in the *ApoE*^−/−^ mice (Figure 5B,C). This is in agreement with a previous study that showed that NR reduces total cholesterol levels in C57BL/6 mice fed a high-fat diet, but different from another study that showed no effect on cholesterol after treatment with nicotinamide in hyperlipidemic *ApoE*^−/−^ mice [32]. Atherosclerotic plaque coverage was reduced in NR-treated mice, indicated by the photo of the aortic arch (Figure 5D, upper panel images), en face Oil Red O staining (Figure 5E), and Oil Red O staining of the aortic root cross-sections (Figure 5D, low panel histology, and Figure 5F). In addition, the EDR of the non-plaque-bearing aortic segment also improved significantly after NR treatment (Figure 5G), while the endothelium-independent vasodilation response to SNP was unaltered (Figure 5H). These results suggested that NR improved endothelial function and alleviated plaque formation in *ApoE*^−/−^ mice.

### 3.6. NR Decreased Macrophage Infiltration and Vascular Inflammation in ApoE^−/−^ Mice

We further examined whether vascular inflammation was also attenuated by NR treatment in these mice by measuring the expressions of the inflammatory factors involved in chemotaxis, cell adhesion, and cytokine responses. Immunofluorescence staining of the aortic root sections showed that NR treatment reduced the expression of E-Selectin (Figure 6A,B), which is the adhesion molecule expressed on activated endothelium to recruit leukocytes in response to cytokines, and reduced the expression of ICAM-1 (Figure 6A,C), which mediates both adhesion and trans-endothelial migration of leukocytes, in the plaque area. Concomitantly, macrophage infiltration in the plaque was also suppressed after NR treatment, as indicated by CD68 staining (Figure 6A,D), which is related to the NR-induced suppression of endothelial activation and subsequent monocyte/macrophage recruitment and infiltration into the plaque. The mRNA expression also demonstrated inhibition of the adhesion molecules Vcam-1, Icam-1, and E-selectin, and the cytokines IL-1β, TNFα, and IL-8 in the aortas collected from *ApoE*^−/−^ mice (Figure 6E). Most of these inflammatory factors are also target genes of NF-κB, which is known to be a target of SIRT3-induced suppression of vascular inflammation. The expression of Nlrp3, another target of SIRT3, was also attenuated after NR treatment (Figure 6E). Moreover, the expression of both Sirt3 or and Sirt1 was unaltered by NR treatment, at least in the vascular tissues. The results from Figure 5 and Figure 6 support the anti-inflammatory and athero-protective effect of NR supplements in mice.

## 4. Discussion

Our present results demonstrate that NAD^+^ attenuates the induction of inflammatory factors, ROS generation, mitochondrial dysfunction, and endothelial dysfunction in ECs induced by inflammatory cytokines. The endogenous expression of SIRT3 is protective against inflammatory responses and mitochondrial ROS generation in ECs. Importantly, NAD^+^ showed both SIRT3-dependent and -independent effects. The NAD^+^-dependent effect of SIRT3 is mostly related to the regulation of NO bioavailability of ECs. We also showed that NR supplementation was able to inhibit plaque growth induced by hyperlipidemia in *ApoE*^−/−^ mice.

In the last part of our study, we showed the in vivo effect of NR in *ApoE*^−/−^ mice. Similarly, in a recent study using *ApoE*^−/−^ mice fed with HCD, it was demonstrated that nicotinamide supplementation in drinking water of approximately 0.5 and 1.9 g/kg/day reduced plaque formation, with an improvement of the lipid profile [33]. The lower dose in this study is similar to that in our study. Although the diet we used contained more cholesterol (1.2% cholesterol), the effect on plaque coverage and vascular inflammation markers were similar. Treatment with another NAD^+^ precursor, nicotinamide mononucleotide (NMN), improved the impaired blood flow and restored vasculature in aging muscle [39]. NMN also upregulated genes related to the anti-atherogenic effects in the aortas of aged mice [40]. In addition, NMN improved NO-dependent vasodilation and attenuated vascular stiffening in old mice with the upregulation of SIRT1 [30]. Although previous studies showed that NR ameliorated metabolic abnormalities in obese mice [18], some more recent studies also showed that NR had minimal effect in various disease models, such as mildly obese mice [41], or in modifying muscle metabolism and acetyl-proteome [42]. Regarding vascular dysfunction and vascular aging, the majority of these studies using a NAD^+^ precursor treatment focused on aging-related vascular dysfunction, while our result showed that in young mice, NAD^+^ supplementation was able to retard plaque growth with the inhibition of vascular inflammation and endothelial dysfunction.

Regarding the effect of SIRT3 expression on atherosclerosis, SIRT3 downregulation was observed within the atherosclerotic plaques of vascular tissue from mice [43] and in the aortas of atherosclerotic rats [44]. In addition, SIRT3 downregulation seems to also be associated with high lipid levels such as in the liver tissue of fatty liver mouse models or patients with fatty liver disease [19,45,46]. However, in Sirt3 knockout mice in a *Ldlr*^−/−^ background model to induce atherosclerosis, the plaque formation or stability remained unaltered [47]. Importantly, certain drugs exert anti-atherogenic effects through SIRT3. For example, melatonin induces SIRT3 activation in atherosclerotic plaques to ameliorate the progression of atherosclerosis [43]. Idebenone, which is a compound similar to co-enzyme Q10, also increases SIRT3 activity and further activates SOD2, which suppresses ROS to ameliorate atherosclerosis in *ApoE*^−/−^ mice [48]. Metformin, the widely used antidiabetic drug, also increases both SIRT1 and SIRT3 [49]. In addition, many nutraceutical and herbal components, such as resveratrol, also upregulate SIRT3, demonstrating anti-aging potential [50]. In summary, the anti-atherogenic effect of NAD^+^ supplementation and the effect of SIRT3 activation in alleviating atherosclerotic plaque formation as reported in the literature are in accordance with our current results. 

In addition to the anti-atherogenic effect of NR, we also demonstrated the anti-inflammatory and anti-oxidative effects of NAD^+^ in ECs. We showed that the endogenous expression of SIRT3 is protective against IL-1β-induced effects in ECs. This is in accordance with a previous report that showed that the depletion of Sirt3 promotes vascular inflammation, while increased Sirt3 expression and activity inhibits the expression of inflammatory markers and reduces inflammatory cytokine levels [51]. It is possible that NAD^+^ and its precursors exert an anti-inflammatory effect through molecules other than SIRT3, such as SIRT1, PARPs, and ectoenzymes. In addition, the antioxidant effect of SIRT3 through the activation of several mitochondrial antioxidant enzymes, such as SOD2, by deacetylation was also observed in our study. The antioxidant effect of SIRT3 may also help to suppress inflammatory responses in endothelial cells [52]. Importantly, in conditions such as metabolic syndrome and aging, with dysregulated or downregulated SIRT3, supplementation with NAD^+^ and its precursors, such as NR, could still be beneficial to reverse vascular pathology. Interestingly, we also found that the effect of NAD^+^ to improve endothelium-dependent vasodilation was abolished with the endothelium-selective deletion of Sirt3. This might be due to the regulation of some enzymes or modulators involved in the NO pathway in ECs regulated by SIRT3, which requires further investigation.

## 5. Conclusions

In summary, we showed that increasing the NAD^+^ level or its biosynthesis attenuates vascular inflammation, oxidative stress, endothelial dysfunction, and atherosclerosis in endothelial cells and in hyperlipidemic *ApoE*^−/−^ mice. In addition, the effect of NAD^+^ is partially mediated by SIRT3, suggesting that NAD^+^ is beneficial against vascular dysfunction in conditions with SIRT3 inhibition.

## Figures and Tables

**Figure 1 antioxidants-11-00706-f001:**
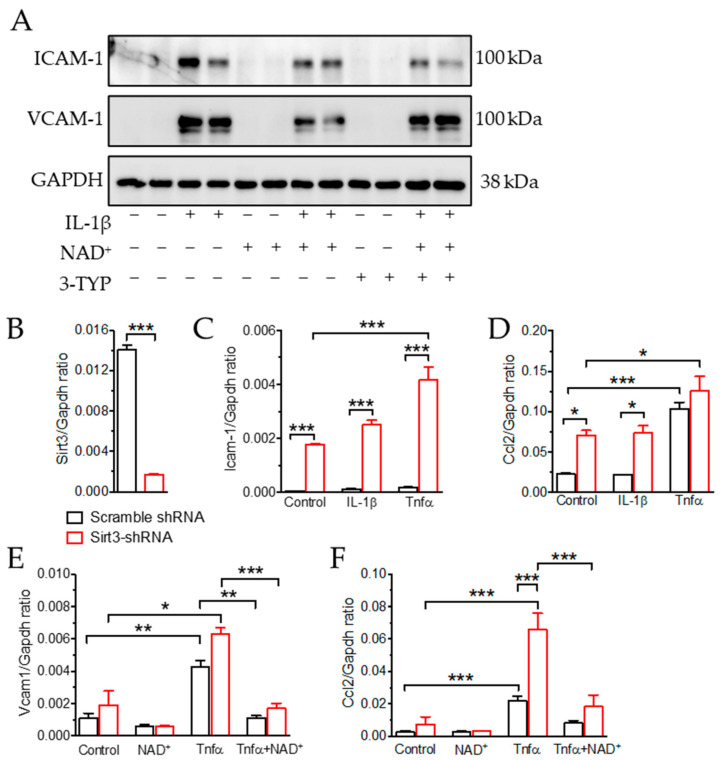
NAD^+^ attenuated inflammatory response in endothelial cells. (**A**) Representative Western blot showing the expression of adhesion molecules VCAM-1 and ICAM-1 in HUVECs treated with 10 ng/mL of IL-1β with or without 1 mmol/L NAD^+^, 100 µmol/L 3-Typ for 16 h. +: in the presence of the indicated treatment; −: without the indicated treatment. (**B**) mRNA of Sirt3 to confirm the efficacy of shRNA knockdown in mBMECs. (**C**,**D**) mRNA expression of Icam1 and Ccl2 in mBMECs with stable expression of either scrambled (Scr) or Sirt3 shRNA (sh-Sirt3) with or without 10 ng/mL of TNFα or 10 ng/mL of IL-1β treatment for 16 h. (**E**,**F**) mRNA expression of Vcam1 and Ccl2 in mBMECs with stable expression of either scrambled (Scr) or Sirt3 shRNA (sh-Sirt3) treated with 10 ng/mL of TNFα with or without 1 mmol/L NAD^+^ for 16 h. Data are means ± SEM. * *p* < 0.05, ** *p* < 0.01, *** *p* < 0.001 between groups.

**Figure 2 antioxidants-11-00706-f002:**
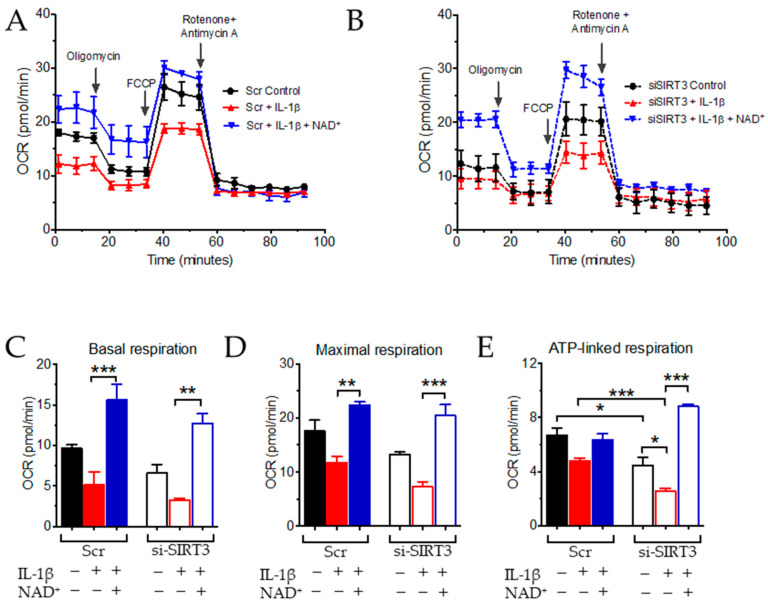
NAD^+^ improved mitochondrial respiration in endothelial cells. (**A**) Oxygen consumption rate (OCR) measured by Seahorse XF Cell Mito Stress Test Kit in HUVECs transfected with scramble (Scr) (**A**) or siSIRT3 (**B**) after treatment with 10 ng/mL of IL-1β and 1 mmol/L NAD^+^ for 16 h. (**C**–**E**) Quantification of baseline respiration (**C**), maximal respiration (**D**) and ATP production (**E**). *n* = 3 experiments. Data are means ± SEM. * *p* < 0.05, ** *p* < 0.01, *** *p* < 0.001 between groups. +: in the presence of the indicated treatment; −: without the indicated treatment.

**Figure 3 antioxidants-11-00706-f003:**
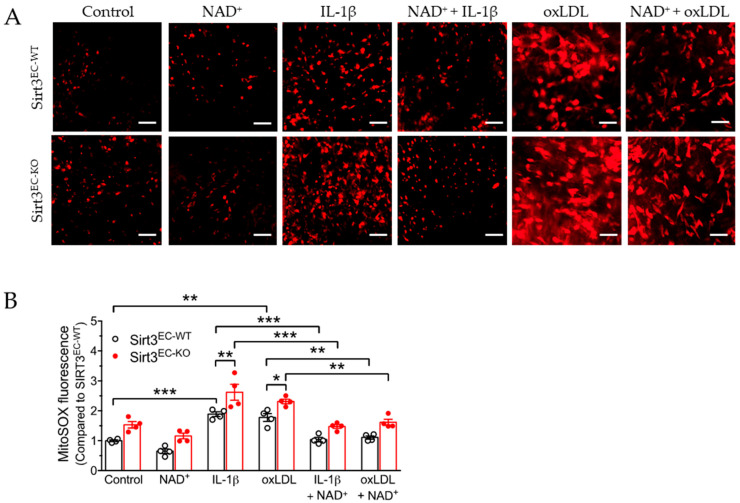
NAD^+^ attenuated oxidative stress in the endothelium. (**A**) Representative images of en face preparation for MitoSOX fluorescence in aortic endothelium isolated from Sirt3^EC-WT^ and Sirt3^EC-KO^ mice treated with 10 ng/mL of IL-1β, or 50 µg/mL of oxLDL with or without 1 mmol/L of NAD^+^ for 16 h. Scale bar = 200 µm. (**B**) Quantification of MitoSOX fluorescence. *n* = 4 mice in each group. Data are means ± SEM. * *p* < 0.05, ** *p* < 0.01, *** *p* < 0.001 between groups.

**Figure 4 antioxidants-11-00706-f004:**
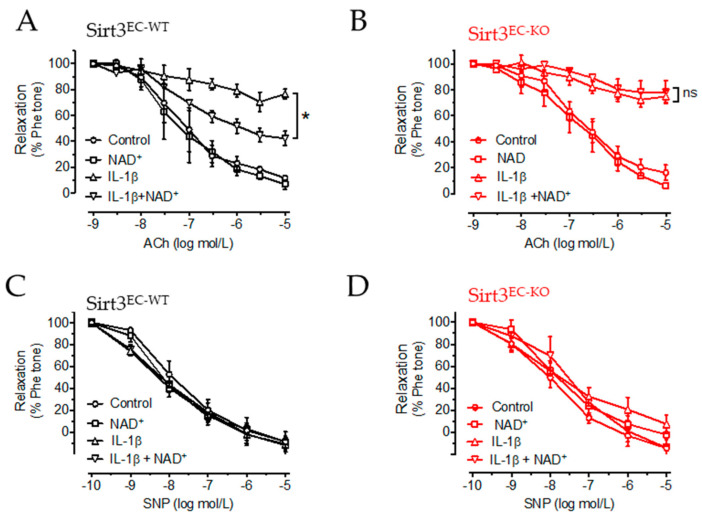
NAD^+^ improved vascular function. (**A**,**B**) Endothelium-dependent relaxation (EDR) of aortic rings from Sirt3^EC-WT^ mice (**A**) and Sirt3^EC-KO^ mice (**B**) in response to acetylcholine (ACh). (**C**,**D**) SNP-induced endothelium-independent relaxation in aortas of Sirt3^EC-WT^ mice (**C**) and Sirt3^EC-KO^ mice (**D**). *n* = 5 mice in each group. Data are means ± SEM. * *p* < 0.05 of AUC between groups.

**Figure 5 antioxidants-11-00706-f005:**
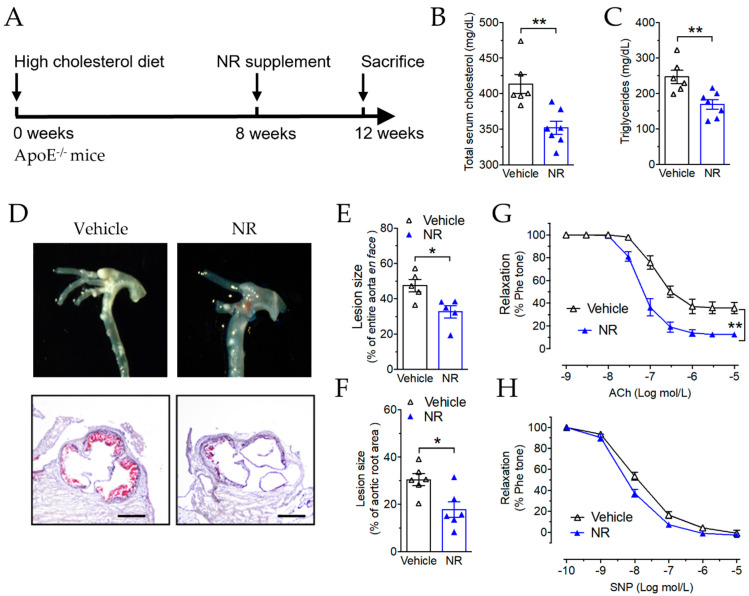
Nicotinamide riboside (NR) attenuates high-cholesterol diet-induced atherosclerosis in *ApoE*^−/−^ mice. (**A**) Schematic experimental outline for NR treatment on *ApoE*^−/−^ mice. *ApoE*^−/−^ mice were fed a high-cholesterol diet for 8 weeks and then were randomly divided into two groups, i.e., the vehicle (water) and NR (400 mg/kg/day), and were fed through oral gavage for another 4 weeks. Mice were sacrificed at 12 weeks and atherosclerotic plaques were quantified. (**B**,**C**) Total cholesterol (**B**) and triglyceride levels in the plasma from *ApoE*^−/−^ mice treated with vehicle or NR. (**D**–**F**) Representative images (**D**) and quantification of lipid deposition in the whole aorta (**E**) and the aortic root cross-section (**F**) were measured by Oil Red O staining. Scale bar = 500 μm. (**G**,**H**) EDR (**G**) and SNP-induced relaxation (**H**) in aortas. *n* = 5 mice per group. Data are means ± SEM. * *p* < 0.05, ** *p* < 0.01 between groups.

**Figure 6 antioxidants-11-00706-f006:**
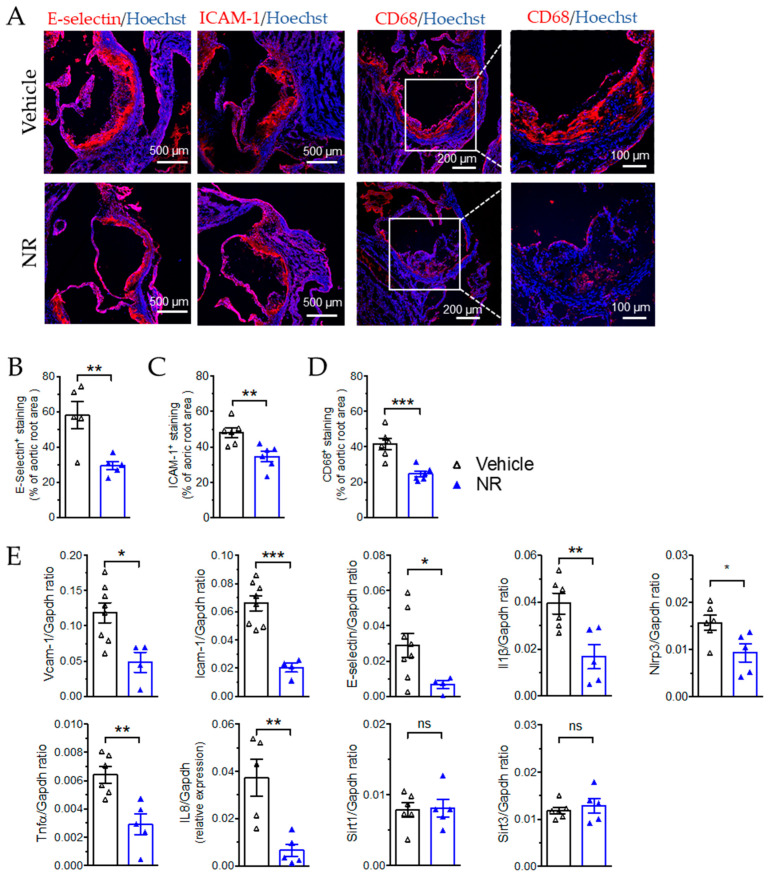
Nicotinamide riboside (NR) attenuated macrophage infiltration and vascular inflammation in *ApoE*^−/−^ mice. (**A**) Immunofluorescence staining of E-selectin and ICAM-1 (endothelial activation markers), and CD68 (macrophage marker) in frozen aortic root sections. Scale bar = 500 µm. *n* = 6. (**B**–**D**) Quantification of the immunofluorescence staining E-selectin (**B**), ICAM-1 (**C**), and CD68 (**D**). (**E**) qRT-PCR result showing mRNA expression of genes involved in vascular inflammation, including Vcam-1, Icam-1, E-Selectin, IL-1β, Tnfα, IL-8, Nlrp3, and Sirt1 and Sirt3 in the aortas from *ApoE*^−/−^ mice treated with vehicle or NR. *n* = 5–7 per group. Data are means ± SEM. * *p* < 0.05, ** *p* < 0.01, *** *p* < 0.001 between groups.

**Table 1 antioxidants-11-00706-t001:** List of mouse primers used for RT-PCR.

Accession Number	mRNA	Forward (5′-3′)	Reverse (5′-3′)
NM_011333	Ccl2	CATCCACGTGTTGGCTCA	GATCATCTTGCTGGTGAATGAGT
NM_011345	E-selectin	AGTTGTGAGTTCTCCTGCGA	CACTCCATGACGCCATTCTG
NM_001289726	Gapdh	ATGGTGAAGGTCGGTGTGAA	GAGGTCAATGAAGGGGTCGT
NM_010493	Icam1	AAACCAGACCCTGGAACTGCAC	GCCTGGCATTTCAGAGTCTGCT
NM_008361	IL-1β	GAAATGCCACCTTTTGACAGTG	TGGATGCTCTCATCAGGACAG
NM_011339	IL-8	GGTGATATTCGAGACCATTTACTG	GCCAACAGTAGCCTTCACCCAT
NM_019812	Sirt1	GCTGACGACTTCGACGACG	TCGGTCAACAGGAGGTTGTCT
NM_022433	Sirt3	ATCCCGGACTTCAGATCCCC	CAACATGAAAAAGGGCTTGGG
NM_001278601	Tnfα	CAGCCTCTTCTCATTCCTGC	ATGAGAGGGAGGCCATTTG
NM_011693	Vcam1	ACAGACAGTCCCCTCAATGG	TCCTCAAAACCCACAGAGCT
NM_145827	Nlrp3	GTGTGGATCTTTGCTGCGAT	TATCCCAGCAAACCCATCCA

## Data Availability

The data presented in this study are available in this manuscript.

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
