# Peer review of "Sirtuin 3 Dependent and Independent Effects of NAD+ to Suppress Vascular Inflammation and Improve Endothelial Function in Mice"

_antioxidants, 2022, doi:10.3390/antiox11040706_

Round 1

Reviewer 1 Report

The manuscript “Sirtuin 3 dependent and independent effects of NAD+ to suppress vascular inflammation and improve endothelial function in mice” is a research article that explored the effects associated to a  endothelial-selective SIRT3 deletion upon vascular inflammation, and the impact of NAD+ to revert it in both endothelial cells and mouse model.  Several grammar mistakes are present (some of them specified below) and the construction of the sentences is sometimes not clear. Therefore a revision of the language is suggested. Nonetheless, the findings of this study are interesting and elucidate some aspects of the theme.   Therefore, the manuscript could be considered for publication answering to the following concerns:

Major

  1. Cell culture: Please specify FBS manufacturer as well as concentration and manufacturer of antibiotics.
  2. Line 88: “In the introduction section authors state that supplementation of nicotinamide riboside (NR), the precursor of NMN, increases plasma and intracellular NAD+”. It is important to specify that also the extracellular conversion of NMN to nicotinamide riboside represents an important vasoprotective mechanism to maintain intracellular NAD (PMID: 32389638).
  3. Line 131: Authors must provide more details about how cells were selected with puromycin (concentration, replacing frequency…).
  4. Quantitative PCR: was the total RNA examined for purity?
  5. Western blot: “Protein from cells or tissues was lysed in RIPA”. This is uncorrect; authors lysed cell pellets or tissues with lysis buffer, not proteins.
  6. Western blot: please specify the amount of total proteins loaded per well; blocking time; concentrations of primary and secondary antibodies used.
  7. In material and methods section please specify the number of passages of HUVECs used for this study.
  8. Line 93: Authors correctly specify that NAM supplementation also prevents the development of atherosclerosis; they should specify that this occurring is also associated to Nicotinamide N-methyltransferase activity in endothelium, which protects endothelial cells against oxidative stress (PMID: 34153425).
  9. Figure 1: authors showed results of genes expression as ratio gene/reference gene; therefore using the expression “relative expression” on y axis is wrong; it should be replaced by “gene/GADPH ratio”.
  10. Figure 1b: why is there no SEM on Sirt3-shRNA?
  11. In the whole manuscript authors wrote the name of the analyzed genes with italics lowercase. When they refere to the gene it is corrected; however, when they refere to mRNA or protein encoded by a gene, they should write the name in uppercase without italics; please reconcile through the text and figures. For instance, in figure 1A they use both styles independently, creating confusion.
  12. In the discussion section authors mention some drugs which exert an anti-atherogenic effects by suppressing ROS. They should specify that oxidative stress is highly detrimental for endothelium since it impairs also mitochondrial activity and permeability (PMID: 33123312).

Minor

Line 30: “SIRT3 is a NAD+-dependent deacetylase that mainly located in mitochondria”; the sentence lacks of the verb. It should be “SIRT3 is a NAD+-dependent deacetylase that is mainly located in mitochondria”.

Line 31: “It was first identified as yeast SIR2, equivalent to SIRT3, possesses NAD+-dependent protein 32 deacetylase activity”. The sentence is uncorrect and should be rephrased; e.g. “It was first identified as yeast SIR2, equivalent to SIRT3, possessing NAD+-dependent protein deacetylase activity”.

Line 54: Please replace “NAD” by “NAD+

Line 64: “which mostly focused on its inhibition of oxidative stress”; The sentence is uncorrect and should be rephrased; “which is mostly focused on its inhibition of oxidative stress”.

line 66: “Sirt3” is written differently than in other parts of the manuscript; please reconcile.

Author Response

Please see the attached file for point by point response to reviewer's comments.

Reviewer 2 Report

The authors have made a very interesting research work investigating in vitro and vivo the NAD+ dependent and independent effect of Sirt3 on atherosclerosis. Authors have used state-of-the-art techniques, using in vitro genetic silencing experiments and have generated an endothelial-targeted conditional knockout mouse model with Sirt3 ablation in the endothelial cells. Despite the fact that the manuscript is scientifically sound and well supported. Some issues emerge from reviewing the manuscript.

Major Comments

  1. One deficit of the paper is that authors do not report on the lipid profile of the mice. It would be interesting if authors could provide data on Total Cholesterol, LDL and triglycerides both from the conditional knockout mice and the ApoE-/- mice after NR administration. This could also be supplemental to their findings considering the mechanisms of endothelial dysfunction.
  2. The discussion could be expanded. Despite the fact that authors comment on some compounds that increase Sirt3 expression and exert anti-atherosclerotic effects, the authors have omitted to comment on translational approaches on regulating Sirt3. For instance, Metformin, a clinically applicable anti-diabetic drug is known to upregulate Sirt3 expression (PMID: 28870631). This could be elaborated within the discussion.
  3. Some downstream targets of Sirt3 should be further investigated, in order to strengthen the hypothesis. Despite the fact that authors have reported on several inflammation-related molecules in their in vitro and in vivo models they have omitted to investigate Sirt3 impairment-related downstream targets such as NLRP3, NFkB or VE-cadherin that are known to be implicated with Sirt3 impairment-related vascular dysfunction (PMID: 31852393). The authors are advised to investigate these targets on a transcriptional and post-transcriptional level.
  4. Despite the fact that the whole manuscript, addresses atherosclerosis on the basis of vascular inflammation, there is no data on vascular infiltration by immune cells. The manuscript would benefit from additional experiments on vascular infiltration, such as FACs analysis or immunohistochemistry analysis of the aortas from the in vivo models.

Minor Comments

  1. Some linguistic deficits should be amended within the manuscript. For instance, in Figure 5A, in the workflow, the word sacrifice is completely misspelled.
  2. In the animal section, authors have omitted from providing an animal application number. The in vivo experiments should be reported according to the ARRIVE guidelines.

Author Response

(The authors gave the same response as above.)

Round 2

Reviewer 2 Report

My concerns are now addressed by the authors